# Effect of Metformin on T2D-Induced MAM Ca^2+^ Uncoupling and Contractile Dysfunction in an Early Mouse Model of Diabetic HFpEF

**DOI:** 10.3390/ijms23073569

**Published:** 2022-03-25

**Authors:** Maya Dia, Christelle Leon, Stephanie Chanon, Nadia Bendridi, Ludovic Gomez, Jennifer Rieusset, Helene Thibault, Melanie Paillard

**Affiliations:** 1Laboratoire CarMeN—IRIS Team, INSERM, INRA, Université Claude Bernard Lyon-1, INSA-Lyon, Univ-Lyon, 69500 Bron, France; maya.dia94@gmail.com (M.D.); christelle.leon@univ-lyon1.fr (C.L.); ludovic.gomez@univ-lyon1.fr (L.G.); helene.thibault@chu-lyon.fr (H.T.); 2Laboratoire CarMeN—MERISM Team, INSERM, INRA, Université Claude Bernard Lyon-1, INSA-Lyon, Univ-Lyon, 69921 Oullins, France; stephanie.chanon@univ-lyon1.fr (S.C.); nadia_bendridi@hotmail.com (N.B.); jennifer.rieusset@univ-lyon1.fr (J.R.); 3Hospices Civils de Lyon, 69500 Bron, France

**Keywords:** diabetic cardiomyopathy, heart failure, type 2 diabetes, reticulum-mitochondria interactions, Ca^2+^ signaling

## Abstract

Diabetic cardiomyopathy (DCM) is a leading complication in type 2 diabetes patients. Recently, we have shown that the reticulum-mitochondria Ca^2+^ uncoupling is an early and reversible trigger of the cardiac dysfunction in a diet-induced mouse model of DCM. Metformin is a first-line antidiabetic drug with recognized cardioprotective effect in myocardial infarction. Whether metformin could prevent the progression of DCM remains not well understood. We therefore investigated the effect of a chronic 6-week metformin treatment on the reticulum-mitochondria Ca^2+^ coupling and the cardiac function in our high-fat high-sucrose diet (HFHSD) mouse model of DCM. Although metformin rescued the glycemic regulation in the HFHSD mice, it did not preserve the reticulum-mitochondria Ca^2+^ coupling either structurally or functionally. Metformin also did not prevent the progression towards cardiac dysfunction, i.e., cardiac hypertrophy and strain dysfunction. In summary, despite its cardioprotective role, metformin is not sufficient to delay the progression to early DCM.

## 1. Introduction

In diabetic patients, cardiovascular disease is the first cause of mortality. Type 2 diabetes (T2D) is associated with a 2 to 4-fold increase in the development of both types of heart failure (with preserved (HFpEF) or reduced ejection fraction (HFrEF)), independently of other cardiovascular risk factors or the presence of a coronary artery disease (CAD) [1]. This suggests a specific myocardial alteration in T2D patients called diabetic cardiomyopathy (DCM) [1,2,3]. Clinically, DCM is characterized by functional and structural alterations of the myocardium in the absence of other cardiac risk factors, and its diagnosis remains challenging [4]. A study on a small cohort reported a 16.9% prevalence of pure DCM with higher morbimortality, which could be further enhanced by coronary artery diseases [5]. Therefore, facing the highly rising prevalence of T2D currently worldwide, limiting the evolution of T2D patients towards DCM and HF remains a great challenge for the medical community.

Since the 1950s, metformin has been recognized as a powerful antihyperglycemic drug and is therefore the first-line treatment prescribed to treat T2D patients [6]. Apart from its glucose lowering potential, metformin was shown to be cardioprotective via the activation of the Adenosine monophosphate activated protein kinase (AMPK) pathway [7], notably improving the mitochondrial organization and function [8,9], and also by directly regulating mitochondria and limiting ROS production [10,11]. Indeed, several studies in both T2D patients and animal models have demonstrated the crucial effect of metformin on reducing cell death and infarct size post-myocardial infarction [12,13,14]. Although a meta-analysis of 40 studies reported a benefit of metformin treatment against cardiovascular mortality in T2D patients with CAD, whether metformin could prevent or delay the progression of DCM and notably HFpEF remains not well known and controversial [14,15].

We have recently developed a diet-induced mouse model of early DCM; mice develop cardiometabolic HFpEF, which in many aspects mimics the clinical features in human patients [16]. We have shown that the cardiomyocyte contractile dysfunction is directly associated with a disruption of the functional Ca^2+^ coupling between reticulum and mitochondria (at contact sites called mitochondria-associated reticular membranes or MAMs), notably at the level of the IP3 receptor (IP3R) Ca^2+^ channeling complex [17], further leading to a reduced mitochondrial Ca^2+^ content and altered mitochondrial bioenergetics. Switching back the diabetic mice on a normal diet restores the cardiac insulin signaling, the cardiac reticulum-mitochondria Ca^2+^ coupling, and the contractile function [16]. Altogether, our data unraveled the MAM Ca^2+^ uncoupling as an early but reversible trigger of DCM.

MAMs are indeed recognized as a critical signaling hub inside the cell, with metabolic implications [18]. Interestingly, AMPK has been also shown to play a role in MAM signaling [19,20], suggesting a potential action of metformin at the MAM interface. In this regard, recent studies demonstrated that metformin treatment improves the reticulum-mitochondria Ca^2+^ coupling in the liver of diabetic mice [21,22]. We therefore questioned in this study if metformin could prevent the progression of diet-induced T2D mice towards DCM by preserving the reticulum-mitochondria Ca^2+^ coupling.

## 2. Results

### 2.1. Metformin Partially Rescues the Glycemic Regulation in HFHSD Mice

Five-week-old mice were subjected to a high-fat high-sucrose (HFHSD) or a standard (SD) diet for 16 weeks (Figure 1A), as we have previously shown that 16 weeks of HFHSD trigger DCM [16]. After 10 weeks of diet, the HFHSD mice received a daily oral gavage with either metformin (200 mg/kg, HFHSD + MET) or the vehicle (0.5% methylcellulose) for the last 6 weeks of feeding. Body weight was similar between the HFHSD and the HFHSD + MET groups before the start of the gavage treatment, and significantly higher than the SD mice (HFHSD: 44.3 ± 0.8 g and HFHSD + MET: 42.5 ± 0.9 g versus SD: 28.9 ± 0.6 g, mean ± SEM, *p* < 0.05). At the end of the 16 weeks of feeding, the HFHSD mice, regardless of the drug administration, had similar body weights, being significantly higher than those fed with SD (HFHSD: 42.9 ± 0.9 g and HFHSD + MET: 42.6 ± 1.5 g versus SD: 29.1 ± 0.6 g, mean ± SEM, *p* < 0.05). We next assessed the antidiabetic effect of metformin treatment on both glucose and insulin tolerance tests (Figure 1B,D). Metformin significantly, although partially, improved the glucose and insulin sensitivity of the HFHSD mice, as displayed by the significant reduction of the area under curve for each test (Figure 1C,E). Metformin also reduced the insulinemia level compared to the HFHSD group (*p* = 0.07), to a similar extent as in the SD group (Figure 1F). To further investigate the effect of metformin on insulin signaling at both the hepatic and cardiac levels, AKT phosphorylation on Ser473 was assessed in the liver and heart. At the hepatic level, insulin-induced AKT phosphorylation was significantly reduced in the HFHSD liver compared to the SD group, and metformin treatment tended to partially increase the AKT phosphorylation (Figure 1G, *p* = 0.064). In the heart, whereas 16 weeks of HFHSD led to a significant decrease of AKT phosphorylation upon in vivo insulin stimulation versus the SD mice, metformin did not trigger a significant improvement of insulin-stimulated AKT phosphorylation (Figure 1H, *p* = 0.216), reflecting only a partial rescue of insulin sensitivity after 6 weeks of metformin treatment in the diabetic mouse heart. Altogether, these data confirm the antidiabetic effect of metformin at the systemic level with a minor effect on cardiac insulin resistance.

### 2.2. Metformin Does Not Preserve the Reticulum-Mitochondrial Ca^2+^ Coupling in HFHSD Cardiomyocytes

We have previously shown that the reticulum-mitochondria Ca^2+^ coupling is altered in the HFHSD mice [16]. We thus wondered if metformin could prevent the HFHSD-induced MAM Ca^2+^ uncoupling. Proximity ligation assay between the IP3R and the porin VDAC, both partners of the IP3R Ca^2+^ channeling complex between reticulum and mitochondria [17], revealed a reduced number of proximity points in the HFHSD cardiomyocytes compared to the SD ones, with no improvement by metformin (Figure 2A,B). To further assess if these structural alterations of the MAM Ca^2+^ coupling translate into functional changes, we followed the mitochondrial Ca^2+^ level in freshly isolated cardiomyocytes expressing the FRET sensor 4mtD3cpv by intramyocardial adenoviral injection of the different mice, as previously described [16]. The IP3R-driven Ca^2+^ transfer to mitochondria was studied upon histamine stimulation (Figure 2C). As expected, histamine induced a significantly smaller Ca^2+^ transfer to mitochondria in the HFHSD cardiomyocytes (Figure 2C). However, metformin did not improve the amplitude of the histamine-driven Ca^2+^ transfer to mitochondria (Figure 2D). Our results suggest that metformin treatment for 6 weeks does not preserve the reticulum-mitochondria Ca^2+^ coupling in the diabetic heart.

### 2.3. Metformin Does Not Prevent the Progression towards T2D-Induced Cardiac Dysfunction

We finally questioned if metformin would prevent HFHSD-induced cardiac dysfunction, as previously reported [16]. After 16 weeks of diet, HFHSD mice displayed an increased heart weight, which was not significantly decreased by metformin treatment (HFHSD: 157 ± 5 mg, HFHSD + MET: 150 ± 5 mg, SD: 141 ± 6 mg). Furthermore, echocardiography further revealed a significant increased thickness of the posterior wall in HFHSD hearts versus SD hearts (Figure 3A). Metformin treatment also showed a strong trend towards enhanced posterior wall thickness compared to SD hearts (*p* = 0.078). Calculation of the relative wall thickness further classified both HFHSD and HFHSD + MET hearts as concentrically hypertrophied (mean RWT > 0.42, Figure 3B). Consistent with a phenotype of cardiometabolic HFpEF, both HFHSD and HFHSD + MET mice exhibited a normal fractional shortening while their strain rate function tended to be decreased (Figure 3C,D). Therefore, metformin treatment for 6 weeks does not counteract the HFHSD-induced cardiac dysfunction.

## 3. Discussion

Here we report that 6 weeks of metformin treatment did not prevent the progression of diet-induced T2D mice towards early DCM as it did not preserve the reticulum-mitochondria Ca^2+^ coupling, although being an efficient antidiabetic treatment. Our data demonstrate no effect of metformin on the body weight of our HFHSD + MET mice, in accordance with Silamikele et al. [23]. However, body weight was reported to be decreased by metformin in other studies [24,25]. Interestingly, similar controversial effects of metformin, oscillating between weight-neutral or weight-sparing effect, have been observed in diabetic patients [26]. One could wonder if this differential effect of metformin on body weight could be linked to the dose taken, the frequency of drug administration, and the duration of treatment. In this study, we used a single dose of metformin per day, consistent with a low dose for newly-diagnosed T2D patients (≈1000 mg per day), as previously suggested [24,27]. Whether an oral gavage twice daily or the use of minipump infusion for a longer period would improve the metformin effect has not been tested in this study. Our study also mimics the fact that numerous diabetic patients do not regularly take their antidiabetic medication, and therefore may be under the optimal dose. Additionally, metformin distribution was shown to rely on the organic cation transporter 1 (OCT1) [28], which is highly expressed in the liver and less in the heart [29]. A differential accumulation of metformin between hepatocytes and cardiomyocytes could thus also explain liver rather than cardiomyocyte contribution to metformin antidiabetic effect. Future analyses, notably of the AMPK phosphorylation status, could help decipher the contribution of each organ in the metformin effect.

However, we have recently reported that this metformin dose and delivery were efficient in reducing cell death and infarct size in both in vitro and in vivo mouse models of myocardial infarction [30], supporting an effective metformin treatment. The cardioprotective effect of metformin against ischemia-reperfusion may rely on the previously reported mechanisms, such as increasing nitric oxide availability, decreasing apoptosis, and favoring the adaptation to energy deficiency [7]. Interestingly, the fact that metformin does not prevent the T2D-induced cardiac MAM Ca^2+^ uncoupling could also contribute to its cardioprotective effect against myocardial infarction by counteracting the ischemia-reperfusion-induced increased MAM Ca^2+^ coupling and thus preventing the mitochondrial Ca^2+^ overload [31]. Importantly, although chronic metformin treatment in diabetic patients is linked to reduced myocardial infarct size [13], in these patients, T2D is also independently associated with an increased risk of HF [3]; this supports our results that metformin may not be enough to prevent the progression towards diabetic cardiomyopathy.

We have previously shown that the T2D-induced reticulum-mitochondria Ca^2+^ uncoupling is an early but reversible trigger of cardiac dysfunction [16]. The fact that metformin treatment did not prevent the alteration of the MAM Ca^2+^ coupling in the HFHSD cardiomyocytes, and the T2D-induced cardiac dysfunction, further support the crucial role of the reticulum-mitochondria Ca^2+^ coupling in controlling the excitation-energetics coupling, as a potential therapeutic target in DCM.

In 2021, the reduction of the combined risk of cardiovascular death or hospitalization in HFpEF patients by Empagliflozin, a sodium–glucose cotransporter 2 inhibitor (SGLT2i), was demonstrated in the EMPEROR-preserved trial [32]. In parallel, several studies suggested a potential competitive effect of metformin with HF treatments [33], including sulphonylureas [34] and SGLT2i in the CANVAS [35] and EMPA-REG OUTCOME trials [36]. Recent investigations are now suggesting a direct cardiac effect of gliflozins [37,38]. Further mechanistic studies are therefore required to determine if SGLT2i improves the cardiac function by preserving the MAM Ca^2+^ coupling in models of diabetic HFpEF, and whether metformin could interfere with the protective effect of SGLT2i.

In summary, our study shows that despite its cardioprotective role against myocardial infarction, daily metformin treatment over several weeks is not sufficient to prevent the alteration of the cardiac reticulum-mitochondria Ca^2+^ coupling induced by T2D and to limit the progression of diabetic cardiomyopathy.

## 4. Materials and Methods

### 4.1. In Vivo Animal Experiments

Mice were from the same protocol as previously published [16,30]. At the age of 5 weeks, male C57BL/6JOlaHsd mice were either fed with a high-fat high-sucrose diet (HFHSD: 260HF U8978 version 19, SAFE: 20% proteins, 36% lipids) or a standard diet (SD: LASQC diet Rod16-A, Genobios: 16.9% proteins, 4.3% lipids) for 16 weeks. During the last 6 weeks of the diet, a randomly chosen group of mice under the HFHSD was subjected daily each morning to a metformin oral gavage (200 mg/kg), whereas the other mice were subjected to the vehicle gavage (0.5% methylcellulose). A total of 42 mice was used in this study. Glucose and insulin tolerance tests and in vivo insulin signaling were performed at 16 weeks, as previously described [16]. Echocardiography was performed under a light anesthesia (ketamine 80 mg/kg ip) with a digital ultrasound system (Vivid 7, GE Medical Systems) and a 13-MHz linear-array transducer as previously described [16]. One week before the cardiomyocyte isolation, intramyocardial injection of the 4mtD3cpv adenovirus (5 × 10^8^ PFU) was performed on isoflurane-anesthetized mice, as previously detailed [16].

### 4.2. Ex Vivo Experiments on Isolated Mouse Cardiomyocytes

Cardiomyocyte isolation protocol was performed as similarly stated in our previous paper [16]. Freshly isolated cardiomyocytes were plated and either (1) imaged on a wide-field Leica DMI6000B microscope to measure the mitochondrial Ca^2+^ level upon 10 mM histamine stimulation, or (2) fixed to perform proximity ligation assay between IP3R1 (1/200, sc28614) and VDAC (1/200, ab14734), as previously done [16].

### 4.3. Immunoblot

Frozen hearts were lysed in RIPA buffer as previously described [16]. A quantity of 50 µg of proteins was loaded on SDS gels, then transferred onto a nitrocellulose membrane after migration. Phospho-AKT (Ser473) and AKT protein levels were detected using rabbit anti-phospho-AKT (1/1000; Cell Signaling 4060L) and rabbit anti-AKT (1/1000; Cell Signaling 4691S).

### 4.4. Statistical Analysis

Analysis and graph representation were performed on GraphPad Prism 9.2.0 (GraphPad Software, San Diego, CA, USA). Statistical analysis is detailed in each figure legend. Non-parametric tests were performed for groups with less than four animals, and were expressed as median. Otherwise, after validation of normality and homoscedasticity, parametric tests were done and data expressed as mean.

## Figures and Tables

**Figure 1 ijms-23-03569-f001:**
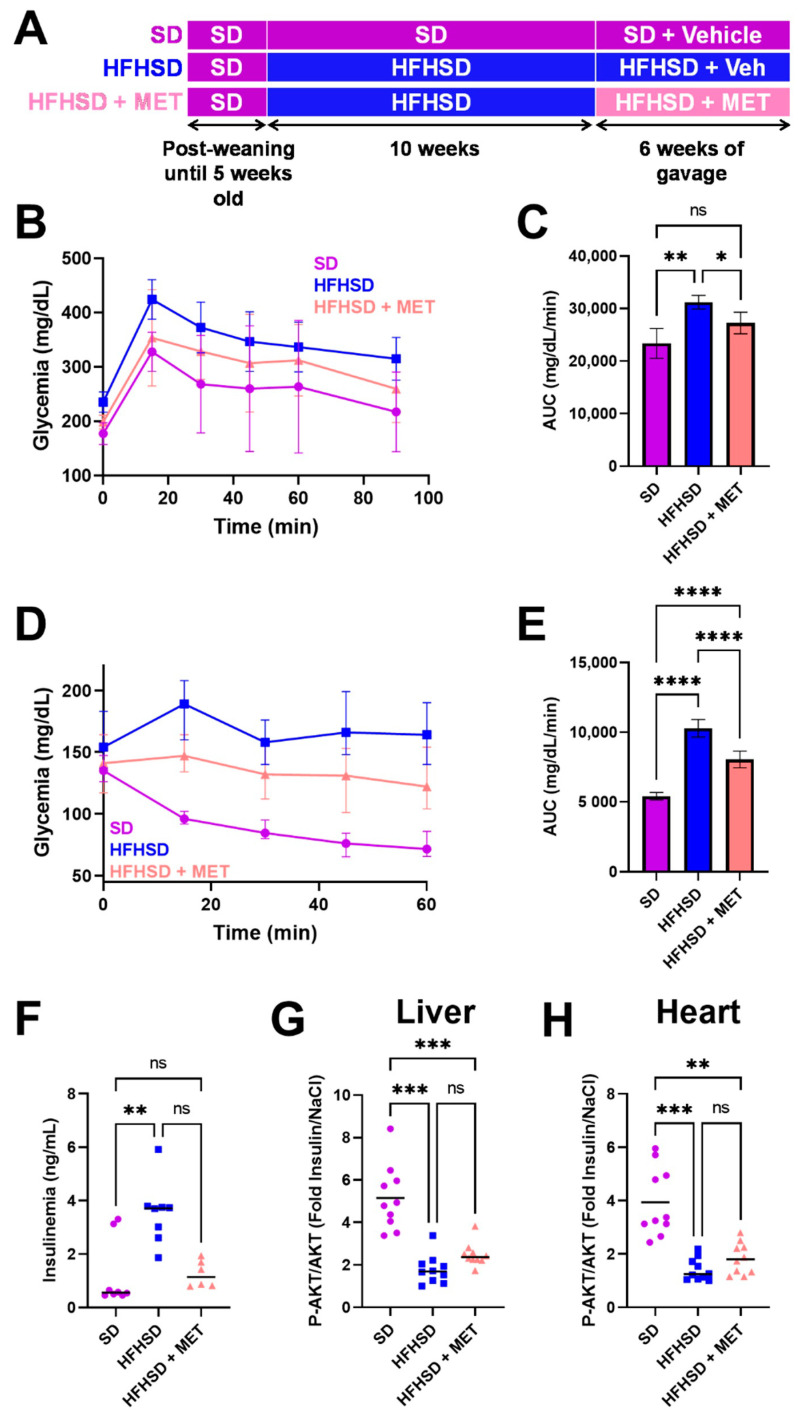
Evaluation of metformin effects on the metabolic status of the HFHSD mice. (**A**) Timeline of the experimental protocol. Daily oral gavage consisted on either metformin (200 mg/kg, HFHSD + MET) or the vehicle (0.5% methylcellulose). Measurements of glycemia following a glucose (**B**) or insulin (**D**) tolerance test, and respective quantification of the area under curve (AUC) in (**C**,**E**) (n = 5 mice/group for GTT and 14–15 mice/group for ITT). Mean ± SD, Brown–Forsythe, and Welch ANOVA test. (**F**) Quantification of blood insulin level. (**G**,**H**) Analysis of insulin sensitivity in liver (**G**) and heart (**H**) by quantification of the cardiac phosphorylation of AKT (on Ser473), calculated as a fold increase of insulin-induced AKT phosphorylation over NaCl (n = 10 mice/group). Mean, Brown–Forsythe, and Welch ANOVA test. * *p* < 0.05, ** *p* < 0.005, *** *p* < 0.001, **** *p* < 0.0001. ns = non-significant. Symbols: circle: SD, square: HFHSD, triangle: HFHSD + MET.

**Figure 2 ijms-23-03569-f002:**
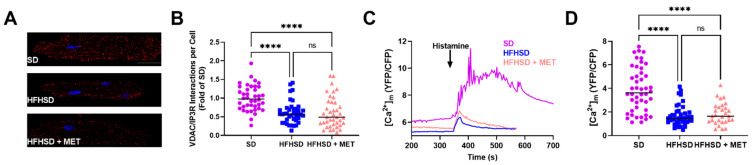
Effect of metformin on the structural and functional MAM Ca^2+^ coupling in the HFHSD cardiomyocyte. (**A**) Representative images of proximity ligation assay between IP3R and VDAC in isolated cardiomyocytes. Blue: nuclei by DAPI. Red dot: proximity between IP3R and VDAC. Scale bar: 25 µm. (**B**) Quantification of the number of interactions between VDAC and IP3R by proximity ligation assay, expressed as a fold of SD; n = 4 mice/group with 10 cardiomyocytes/mouse. (**C**) Representative traces of the mitochondrial Ca^2+^ level ([Ca^2+^]_m_) expressed as a YFP/CFP ratio, with stimulation by 10 mM histamine. (**D**) Quantification of the [Ca^2+^]_m_ amplitude of the histamine-induced peak; n = 4 mice/group with 46 SD, 42 HFHSD, and 29 HFHSD + MET cardiomyocytes. Median, Kruskal–Wallis test. **** *p* < 0.0001. ns = non-significant. Symbols: circle: SD, square: HFHSD, triangle: HFHSD + MET.

**Figure 3 ijms-23-03569-f003:**
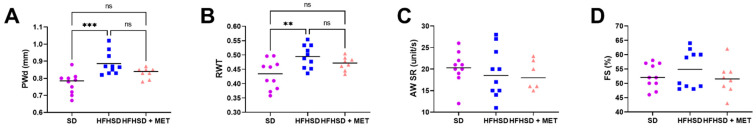
Determination of the metformin effect on cardiac function after 16 weeks of HFHSD by echocardiography. (**A**) Quantification of the posterior wall thickness in diastole (PWd). (**B**) Calculation of the relative wall thickness (RWT = 2 × PWd/LVEDD) as an index of concentric hypertrophy (RWT > 0.42). Quantification of the anterior wall strain rate, AW SR, in (**C**) and of the fractional shortening, FS, in (**D**); n = 8 to 10 mice/group. Mean, one-way ANOVA with Tukey’s multiple comparisons test. ** *p* < 0.005, *** *p* < 0.001. ns = non-significant. Symbols: circle: SD, square: HFHSD, triangle: HFHSD + MET.

## Data Availability

Data supporting the reported results are available on request from the corresponding author.

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
