# Peer review of "Effect of Metformin on T2D-Induced MAM Ca2+ Uncoupling and Contractile Dysfunction in an Early Mouse Model of Diabetic HFpEF"

_ijms, 2022, doi:10.3390/ijms23073569_

Round 1

Reviewer 1 Report

In the present short manuscript Dia et al. have investigated the impact of chronic treatment with the antidiabetic drug metformin on the reticulum-mitochondria calcium coupling in cardiomyocytes from high-fat high-sucrose (HFHS)-fed mice. They mainly showed that, despite a beneficial effect on whole-body glucose homeostasis and insulin sensitivity, metformin neither prevented the alteration of ER-mitochondria calcium coupling in cardiomyocytes nor protected against HFHS-induced diabetic cardiomyopathy. Altogether, the data presented mostly invalidate the authors’ initial hypothesis but are technically sound and scientifically solid, allowing to answer a clear and important question. I have only few minor points/suggestions that might eventually contribute to improve the manuscript.

Minor points

  1. In the introduction, the authors highlighted the role of AMPK in the cardioprotective effect(s) of metformin. However, not all the putative mechanisms underlying cardioprotection by metformin that have been reported/suggested in the literature are linked to AMPK activation by the drug. The direct effect of metformin on mitochondria and the subsequent reduction of hyperglycemia-induced ROS generation / oxidative stress-induced cell death is also one of the possible mechanism. I’d suggest refining a bit this paragraph.
  2. Altogether, metformin does not accumulate so much in cardiomyocytes (low expression of OCT1) by contrast to enterocytes and hepatocytes, probably explaining the absence of effect in this cell type. Maybe something to introduce in the discussion, together with the paragraph on dosage/timing of drug administration ?
  3. If the data are easily available, I’d suggest to add the pPKB/PKB in the liver and the pThr172-AMPK/AMPK in both organs on Figure 1. This will provide solid argument to support/infirm some of your conclusion (liver rather than cardiomyocyte contribution to whole-body IR, tissue-specific AMPK activation)
  4. I’d love to see an illustrative image for VDAC/IP3R PLA in Fig 2
  5. Please indicate in the M&M the residue targeted by the phospho-PKB antibody used (Ser473 or Thr308)

Author Response

We thank the reviewer for the comments and suggestions.

  1. We agree that we were restrictive and we have now mentioned the direct role of metformin on mitochondria and ROS (line 45):

“Apart from its glucose lowering potential, metformin was shown to be cardioprotective via the activation of the AMPK pathway 6, notably improving the mitochondrial organization and function 7, and also by directly regulating mitochondria and limiting ROS production 8.”

  1. Thanks for this interesting suggestion on OCT1 expression which is now discussed (line 180):

“Additionally, metformin distribution was shown to rely on the organic cation transporter 1 (OCT1) 20 which is highly expressed in the liver and less in the heart 21. A differential accumulation of metformin between hepatocytes and cardiomyocytes could thus also explain the liver rather than cardiomyocyte contribution to metformin antidiabetic effect.”

  1. Quantification of P-PKB/PKB after insulin stimulation in the liver of the different mouse groups is now presented in Figure 1 and in the results section (line 79).

“To further investigate the effect of metformin on insulin signaling at both hepatic and cardiac level, AKT phosphorylation on Ser473 was assessed in liver and heart. At the hepatic level, insulin-induced AKT phosphorylation was significantly reduced in the HFHSD liver compared to SD group and metformin treatment tended to partially increase the AKT phosphorylation (Figure 1G, p=0.064).” Unfortunately, we don’t have the P-AMPK/AMPK antibodies, but appreciate the suggestion that is now mentioned in the discussion (line 183).

“Future analyses, notably of the AMPK phosphorylation status, could help decipher the contribution of each organ in the metformin effect.”

  1. Representative images have now been added in Figure 2.

5. It is now specified in the methods that the P-AKT is on Ser473.

Reviewer 2 Report

The paper “Effect of Metformin on T2D-induced MAM Ca2+ uncoupling 2 and contractile dysfunction in an early mouse model of dia- 3 betic HFpEF" by Maya Diaet al. is a murine study with the aim of assessing a potential role of metformin on MAM Ca2+ and contractile disfunction.

The article is well written and only minor spell check is necessary. The paper has a good design. The article is logically divided into sections and subsections. The work has a good degree of novelty and of good interest to the readers.

Comments:

  • Introduction: It should be better explained the role of diabetic cardiomyopathy, in particular, “In the current clinical practice, DM-CMP diagnosis is still challenging, as it requires the identification of distinct functional and structural changes in the LV and the concomitant exclusion of other cardiac diseases and risk factors for CVD. Due to the very frequent confounding of other HF risk factors such as hypertension, CAD, and renal disease, the burden of a “pure” diabetic cardiomyopathy is conceivable not as high as the cardiomyopathy of heterogeneous etiology, with a calculated prevalence of 16.9% of diabetic patients in a small study” (DOI: 3389/fmed.2021.695792).
  • Discussion: A potential role of SGLT2is has been suggested. In a previous letter in the same journal, it has been reported as follows, which could be of help to provide more details: “In diabetic individuals, the increased sodium intracellular concentration is exposed to a much higher risk. In this context, the sodium–hydrogen exchanger 1 (NHE1) and SGLT1 result upregulated, with a consequent significant increase in the intra-cytosolic sodium content. Empagliflozin demonstrated lower cardiac intracellular Na+ and Ca2+, with a higher concentration of mitochondrial Ca2+ in rabbits and rats. Such an effect is mediated by the inhibition of NHE1. Further studies on Dapagliflozin and Canagliflozin in mice proved NHE1 inhibition as a class effect. SGLT2 receptors, not expressed in the heart, better render how NHE1 inhibition is performed. In addition, the inhibition of NHE and subsequent lowering of cardiac cytosolic Na+ seems a potential class effect of SGLT2i to face Hf. NHE1 and NHE3 inhibition may represent a good therapeutic tool to prevent cardiac remodelling and HF. Of note, although the benefits of NHE1 inhibition have been largely demonstrated in experimental models, several studies with NHE-1 inhibitors have not obtained positive results. Anyway, all these observations suggest a key rule for ionic homeostasis in the cardioprotective effects of gliflozins.” (DOI: 3390/ijms22115863).

Author Response

We thank the reviewer for the comments and suggestions.

Point 1: We have now better specified the contribution of DCM in the introduction (line 36):

Clinically, DCM is characterized by functional and structural alterations of the myocardium in the absence of other cardiac risk factors, and its diagnosis remains challenging 3. A study on a small cohort reported a 16.9% prevalence of pure DCM with higher morbimortality, which could be further enhanced by coronary artery diseases 4.

Point 2: We appreciate the reviewer’s suggestion and have now added this point in the discussion (line 202).

Recent investigations are now suggesting a direct cardiac effect of gliflozins 29. Further mechanistic studies are therefore required to determine if SGLT2i improves the cardiac function by preserving the MAM Ca2+ coupling in models of diabetic HFpEF, and whether metformin could interfere with the protective effect of SGLT2i.

Reviewer 3 Report

This is an interesting communication, providing novel evidence on the impact of metformin on T2D induced MAM Ca2+ uncoupling 2 and contractile dysfunction in an early mouse model of diabetic HFpEF. The introduction is sufficient whilst the methods are clearly described. However, questions still persist as to the absence of comparative drug/agent that could be effectively used as a control for this specific study. Perhaps, the choice for the dose/duration or mode of administration could have limited the effects of metformin (discuss)? Nonetheless, the manuscript merits publication since it provides an avenue for further explore the topic. Please indicate future direction, and clarify on what is considered “chronic” treatment in this kind of study?

Author Response

We thank the reviewer for the comments and suggestions.

We have now better discussed the effect of metformin treatment (dose, duration and administration) (line 177). However, with the same daily gavage metformin treatment, we have shown a cardioprotective effect against myocardial infarction in HFHSD mice. Therefore, it seems that metformin is not as potent against the evolution of contractile dysfunction as in reducing myocardial infarction.

“Whether an oral gavage twice daily or the use of minipump infusion for a longer period would improve the metformin effect has not been tested in this study.”

Here we refer to “chronic” treatment when performed daily over several weeks. This has now been specified in the discussion (line 206).

“In summary, our study unravels that daily metformin treatment over several weeks is not sufficient to prevent the alteration of the cardiac reticulum-mitochondria Ca2+ coupling induced by T2D and to limit the progression of diabetic cardiomyopathy.”

Future direction is indicated in the discussion, i.e. 1) deciphering the contribution of liver and heart to the metformin effect (line183) and 2) studying the effect of gliflozins on the MAM Ca2+ coupling during diabetic HFpEF, and the potential interference with metformin treatment (line 202).

“Future analyses, notably of the AMPK phosphorylation status, could help decipher the contribution of each organ in the metformin effect.”

“Recent investigations are now suggesting a direct cardiac effect of gliflozins 29. Further mechanistic studies are therefore required to determine if SGLT2i improves the cardiac function by preserving the MAM Ca2+ coupling in models of diabetic HFpEF, and whether metformin could interfere with the protective effect of SGLT2i.”